# Experimental Study on the Influence of Transverse Crack on Chloride Ingress in Concrete Slab Track of High-Speed Railway

**DOI:** 10.3390/ma16093524

**Published:** 2023-05-04

**Authors:** Xiaochun Liu, Haihua Li, Min Qi, Yiyi Yang, Zhihui Zhu, Zhiwu Yu

**Affiliations:** 1National Engineering Research Center for High Speed Railway Construction, Central South University, Changsha 410075, China; 2School of Civil Engineering, Central South University, Changsha 410075, China

**Keywords:** concrete slab, slab track, chloride diffusion, transverse crack, drying-wetting cycles, durability

## Abstract

The concrete track slab and the base slab of the high-speed railway CRTS II track structure are prone to transverse cracks in the initial service period, which are subjected to environmental action and train load. In order to investigate the influence of transverse cracks on chloride ingress of concrete track slab and base slab in a coastal environment, drying-wetting cycle chloride erosion tests were carried out on reinforced concrete track slab and base slab specimens with cracks ranging from 0 mm to 0.6 mm, subjected to continuous bending moment. The chloride ion concentration of concrete along the depth direction was collected during the test process. The experimental results show that the chloride ion concentration of concrete at the crack section is much higher than that at the intact section, and it increases with the increase of crack width in the range of 0.2 mm to 0.6 mm. A chloride diffusion coefficient model of cracked concrete is proposed for slab track based on the experimental results, which can comprehensively consider the effects of surface chloride ion concentration, chloride binding effect, time-varying effect, temperature, relative humidity, and transverse crack width.

## 1. Introduction

CRTS II slab track structure is a kind of longitudinal continuous ballastless track structure most widely used in high-speed railways in China. It is composed of rails, fasteners, transverse prestressed concrete track slab, cement emulsified asphalt mortar filling layer, concrete base slab, or bearing layer from top to bottom. The concrete track slab and base slab are significant load-bearing components in the track structure, which will guarantee the safety and ride quality of the track structure during operation. However, a large number of engineering practices indicate that the concrete track slab and the base slab of the CRTS II slab track structure are commonly prone to transverse cracks in the initial service period (as shown in Figure 1) due to the combined action of temperature [1,2,3], train load [4,5,6], concrete shrinkage [7,8], and uneven settlement of foundation [9,10]. This kind of transverse cracks could provide a fast channel for corrosive media, moisture, and oxygen [11,12,13,14,15], threatening the durability of concrete slab track structures used in outdoor environments. The durability of CRTS II slab track structures built in the coastal region will face more severe challenges.

Chloride ingress-induced corrosion of steel reinforcement is one of the main reasons for the deterioration of RC structures in coastal environments [16]. Once the chloride ion penetrates into concrete from the outside and reaches the surface of the steel bar, it breaks the passive film of steel rebar and causes corrosion, which leads to the cracking and spalling of the concrete cover and even seriously affects the durability and safety of the structure [17]. Many scholars have carried out a significant number of experimental and theoretical research on the effect of cracks on chloride transmission in concrete. Crack width is one of the main factors affecting chloride transport in concrete [18,19]. Jang SY et al. proposed the concept of “threshold crack width” through steady-state migration tests, and it was concluded that the diffusion coefficient increased with the increase of crack width only when the crack width was greater than the critical value [20]. The literature [21] also supports the view of threshold crack width, and it is believed that the effective diffusion coefficient of chloride ion in the mortar increases rapidly while the crack width is greater than 135 μm. In addition to the crack width, other parameters of the crack, such as orientation [22], density [23], and roughness [11], also have impacts on chloride penetration. Some previous studies found that the existence of cracks will accelerate the ingress of chloride ions in concrete through field investigation [17,24], drying–wetting cycles test [25,26,27,28], non-steady state migration method [22,29,30,31], or rapid chloride migration (RCM) test [32,33]. It was also found that the chloride diffusion coefficient in the vicinity of the crack was larger than that in sound concrete and increased with the increase of crack width. In addition, the relationship between chloride diffusion coefficient and crack width can be expressed as a quadratic function [17,25] or cubic function [34]. In order to more accurately predict the distribution of chloride ions in cracked concrete under actual conditions, several references propose that the apparent chloride coefficient in cracked concrete is not only related to crack width but also influenced by many factors, such as water-cement ratio [35], ambient temperature and relative humidity [36], chloride binding capability [37] and time-varying effect [38]. For the transport model of chloride ions in cracked concrete, Ye et al. proposed a model for the penetration of chloride ions into sound and cracked concrete under drying and wetting cycles and described the distribution of chloride ions in concrete by simplifying and modifying Fick’s second law [11]. Takewaka et al. constructed a simulation model for the deterioration of concrete structures due to chloride attack by considering concrete damage such as cracks, which combined chloride and oxygen penetration models and a reinforcement corrosion model [39]. Park et al. considered a REV model with cracks and proposed a model of chloride ions penetration and diffusion in an unsaturated state according to the chloride ion diffusion test results under unsteady conditions [34].

The diffusion of chloride ion in cracked concrete is a complex physical process. A lot of experimental, theoretical, and simulative research has been conducted on the effect of cracks on chloride transmission in concrete. However, most of the previous related studies on the transmission of chloride ions in cracked concrete mainly focused on low and medium-strength concrete components and very few studies were aimed at high-performance concrete with a concrete grade higher than C50, such as C55 track slabs. Moreover, transverse V-shaped grooves were introduced on the top surface of the track slab to control the position of possible cracks, making the diffusion of chloride ions much more complex in concrete slab tracks. Therefore, the influence of transverse cracks on chloride ingress in concrete slab tracks needs to be further studied.

Aiming at the process of chloride ion ingress in cracked track slab and base slab in a coastal environment, reinforced concrete specimens were designed and prepared according to the details of a track slab with a V-shaped groove and base slab in practical engineering, respectively. A new type of self-balanced continuous loading device with disc springs and stainless-steel screws was designed and manufactured to form and maintain transverse cracks in tested specimens. The chloride ingress drying-wetting cycle tests were conducted on track slab specimens and base slab specimens with different crack widths, and the influence of transverse crack on chloride ingress in concrete slab track components was analyzed according to the measured concentrations of chloride ions in concrete. The research results can provide a theoretical basis for the durability life evaluation and the maintenance of the track structure.

## 2. Experimental Program

### 2.1. Design of Specimens

The reinforced concrete specimens were designed based on the engineering background of the CRTS II slab track structure applied in the Beijing–Shanghai high-speed railway line. The dimension of the specimens, the reinforcement, and the details are shown in Figure 2. In order to comply with the engineering practice, the segmental test specimens were designed to reflect the local details of the track slab and base slab where transverse cracks appeared (as shown in Figure 1). The depth of the specimens, the steel bar diameter, the concrete cover thickness, and the details of the V-shaped groove were consistent with the full-scale CRTS II slab track structure. The designed concrete strength grades of the track slab and base slab were C55 and C30, respectively. Two groups of specimens, named track slab and base slab, were prepared and tested, with 24 specimens in each group, numbered A, B, C, and D according to the typical crack width.

### 2.2. Materials

The mix proportions of the concrete employed in the track slab and base slab specimens are listed in Table 1. Ordinary Portland cement with a strength grade of 42.5 MPa (produced by Yiyang Shaofeng Cement Co., Ltd., Yiyang, China) was used in the study. The mineral ingredients of Portland cement are given in Table 2. Grade I fly ash and Grade II fly ash (produced by Huadian Changsha Power Generation Co., Ltd., Changsha, China) were used in the track slab and base slab specimens, respectively. Natural medium to coarse river sand from the Xiangjiang River was used as fine aggregate, and the fineness moduli of sand used in the track slab and base slab specimens were 2.9 and 2.7, respectively. Two graded crushed gravel with a maximum particle size of 20 mm and a continuous grading ranging from 5 mm to 25 mm were used as the coarse aggregate for the track slab and base slab specimens, respectively. The polycarboxylate superplasticizer with a water reduction rate of 30% was used in the concrete.

Hot rolled ribbed steel bars with a strength grade of 500 MPa (produced by Jiangsu Yonggang Group Co., Ltd., Zhangjiagang, China) and a nominal diameter of 8 mm, 16 mm, and 20 mm were used as the reinforcement for track slab and base slab specimens, respectively. The average values of yield strength and ultimate strength were 502 MPa and 637 MPa, respectively.

### 2.3. Preparation of Specimens

The concrete track slab and base slab specimens were precast in wooden formwork. A steel mold bent by a steel plate with a thickness of 1 mm was introduced to accurately simulate the details of a V-shaped groove, which was fixed on the bottom of the wooden formwork of track slab specimens with AB glue. The steel bar was fixed in the design position by a fine steel wire locked on the side mold of the specimen. The concrete was poured after the formwork and reinforcement were ready. The track slab specimens were wet steam cured for 6 h after the initial setting of concrete. Then they were cured in saturated lime water for 3 days, and stored outdoors at 20 ± 3 °C for 180 days, according to the maintenance process of the track slab. The base slab specimens were naturally cured at a temperature of 20 ± 3 °C for 28 days. Both side surfaces of the specimens were sealed with epoxy resin after they reached a natural drying state.

### 2.4. Loading and Crack Formation

Preset artificial cracks [23,29,32] and static load-induced cracks [25,40,41] are commonly used crack-introducing methods on concrete members in previous chloride ingress experimental studies. For the former method, it is easy to control crack width, while hard to simulate the crack geometry and the pore structure adjacent to the crack. The static load-induced crack method was selected in this study. Although the crack width was difficult to control accurately using this method, it can objectively reproduce the roughness, tortuosity, and capillary pores at the crack surface, which are very close to cracks that appeared in the engineering practice.

As shown in Figure 3, a self-balanced loading device was designed and fabricated to introduce cracks into the specimens and fulfill the continuous loading during the chloride ingress test. The setups of the loading test on the track slab specimen and base slab specimen are shown in Figure 3a,b, which could induce cracks at the weakest section of the V-shaped groove and at the pure bending section, respectively. After the specimen and loading device were installed, they were laid down on two square timbers. The concrete specimen was supported on the steel roll shaft in the mid-span of the steel box reaction beam, and the tension load was subjected to both ends of the specimen by tightening the nuts on the stainless-steel screws slowly with two wrenches.

As shown in Figure 3c,d, during the loading process, the strain on the surface of the specimen was captured by DH1205 strain gauges (surface resistance strain gauge) and collected by an HBM-840b data acquisition instrument. The crack width was continuously observed by the crack observation instrument (produced by Beijing ZBL Technology Co., Ltd., Beijing, China). The loading progress of the track slab and base slab specimens was controlled by strain readings and the maximum crack width readings. The crack observation instrument measured the maximum crack widths at the bottom of the V-shaped groove of the track slab and the top surface of the base slab specimens when the crack width reached stable about 10 min later at each load step. The step loading stopped when the measured crack width reached the predetermined value. Table 3 gives the properties and maximum crack widths of the tested specimens.

### 2.5. Drying-Wetting Cycles Test

Concrete members were usually immersed in chloride solution [11,12,13,25,42,43] or subjected to drying-wetting cycles [41,44,45,46,47,48] to accelerate the chloride erosion process in existing related research. In order to simulate the deposition and transmission mechanism of chloride ions in concrete track slabs and base slabs used in coastal environments, drying-wetting cycle tests were selected in this research.

Once the crack widths of specimens reached and maintained the expected value, they were carefully transported to the chamber and placed vertically in two columns with cracked surfaces upwards on two square timbers. Drying-wetting cycle tests of chloride ingress were subsequently conducted on these specimens. The drying-wetting cycle test was completed in the multi-functional environmental simulation chamber of the National Engineering Research Center for High Speed Railway Construction. The setup of the drying-wetting cycle test on specimens is shown in Figure 4. The minimum vertical distance between the infrared lamp and the top surface of the specimen was controlled by measuring that the maximum temperature on the top surface of the specimen reached 55 °C during the irradiation drying process.

The chloride ingress dry-wetting cycle tests were designed according to T/CECS 762-2020 [49]. The process of a drying-wetting cycle was as follows: firstly, spray the sodium chloride solution on the top surface of the specimens for 2 h by atomizing the nozzle of the environmental simulating chamber at 20 ± 3 °C; secondly, turn on the infrared lamp auxiliary suspended above the specimens and keep the irradiation drying for 20 h at 55 °C; lastly, turn on the ventilation system of the chamber and maintain the environmental temperature at 20 ± 3 °C for 2 h. Chloride solution with a mass concentration of 5% was used in this test, which is consistent with the chloride ions concentration adopted by many scholars [11,12,14,25,41] and the provisions of relevant specifications. The relative humidity in the chamber varied from 98% to 100% and from 80% to 60% during the spraying stage and the drying process, respectively. Sixty drying-wetting cycles were conducted on the base slab specimens, while ninety drying-wetting cycle tests were conducted on the track slab specimens due to the fact that the higher strength and the denser pore structure of concrete will lead to lower transmission velocity.

### 2.6. Determination of Free Chloride Ions Concentration

The concentration of chloride ions in concrete at different depths was tested according to the method specified in JGJ/T 322-2013 [50] and with reference to ASTM C1152/C1152M-04 (2012) [51]. As shown in Figure 5, when the number of dry-wetting cycles of the specimen reached typical values, namely 30, 60, and 90 for track slab specimens and 20, 40, and 60 for base slab specimens, respectively, the dry-wetting cycle was suspended. The powder samples of concrete were prepared along the depth perpendicular to the exposed surface at the cracked section 1-1 and uncracked section 2-2 using a PF-1100 profile grinding machine (produced by Germann Instruments, Hovedstaden, Denmark). Epoxy mortar was used to promptly seal the holes after sampling, and the subsequent drying-wetting cycle test was continued after the mortar had hardened for 12 h.

The concrete samples were dried in an oven at 105 °C for 2 h. The free chloride concentration of concrete was determined by the solid-liquid extraction method for hardened concrete proposed in JGJ/T 322-2013. The content of water-soluble chloride ions in hardened concrete samples is calculated as follows:
(1)$$W_{Cl^{-}}=\frac{3.545C_{{\mathrm{AgNO}}_{3}}V_{3}}{GV_{2}/V_{1}}$$
where $W_{Cl^{-}}$ is the mass percentage of water-soluble chloride ions in concrete and that in the mortar (%); $C_{{\mathrm{AgNO}}_{3}}$ is the concentration of the standard silver nitrate solution (0.0141 mol/L); $V_{3}$ is the volume of silver nitrate standard solution for titration (mL); G is the mass of mortar sample (g); $V_{2}$ is the volume of filtrate extracted per titration (mL); $V_{1}$ is the volume of distilled water used for soaking sample (mL).

It should be noted that SEM and microscopic tests are commonly used to observe the pore structure and ion transport characteristics of concrete. However, SEM and microscopic tests of porosity and permeability were not conducted in this study. The main reason is that the test specimen is a reinforced concrete component subjected to continuous bending moment. Cutting and sampling for SEM and microscopic tests during the test process will cause damage to the specimen, affecting the stress state of the component and subsequent test progress.

The focus of this study is to investigate the effect of transverse cracks on chloride ingress in concrete track structures before severe steel corrosion. In fact, the durability of steel bars embedded in concrete specimens and the reduction characteristics of structural strength and stiffness were investigated in the subsequent study, and the results of the test and numerical simulation will be published in the next paper.

## 3. Results and Discussion

### 3.1. Chloride Ions Concentration in Concrete

The chloride concentration distributions with different times of drying-wetting cycles of chloride erosion and different depths from the exposed surface of concrete specimens at the cracked section 1-1 and uncracked section 2-2 are shown in Figure 6 and Figure 7 for track slab specimens and base slab specimens, respectively.

It can be seen from Figure 6a and Figure 7a that the water-soluble chloride concentration in concrete at the cracked section 1-1 of the track slab specimen and base slab specimen decreases with the increase of depth from the exposed surface. The chloride concentration-depth curve of the track slab specimen is steeper than that of the base slab specimen. It could be attributed to the denser structure and fewer pores in the C55 concrete of the track slab compared with the C35 concrete of base slab, which adversely affects the transport of chloride ions. The chloride concentration at a certain depth from the exposed surface in the cracked concrete is significantly higher than that in sound concrete. In addition, the chloride concentration tends to increase with the increase of crack width in the range of 0.2 mm to 0.6 mm. It also can be seen that chloride concentration in concrete increases with the increases of drying-wetting cycles of chloride erosion and tends to be stable subsequently.

It can be seen from Figure 6b that the chloride concentration in concrete decreases sharply with the increase of depth at section 2-2 of the track slab specimens. The chloride concentration almost decreases to 0 and remains constant at the track slab specimen measurement points with cover thickness greater than 35 mm during the drying-wetting cycles of chloride erosion. According to Figure 7b, the chloride concentration in concrete at section 2-2 of the base slab specimens decreases slowly with the increase of depth, and influence depth reaches 65 mm during 20–60 drying-wetting cycles of chloride erosion, which is much different from the case of track slab specimens. For a given exposure condition and a certain depth, the chloride concentration in sound cover concrete for track slab specimens increases slightly with the increase of the crack width, while it increases substantially for base slab specimens. It could be attributed to the differences in chloride diffusion coefficients and stress levels on different specimens.

By comparing the test results of chloride concentration in Figure 6a,b, it can be seen that the variations of chloride concentration in concrete with the depth are greatly different at section 1-1 from section 2-2 of the track slab specimens. The chloride concentration in section 2-2 without macro cracks generally agrees well with the curve of the chloride diffusion model in sound concrete, while it obviously deviates from the chloride diffusion model in section 1-1 with macro cracks. It can be explained that the presence and development of a transverse crack may allow chloride to penetrate into the interior of the concrete from the side wall of cracks possible, which changes the one-dimensional diffusion-dominated transmission mode of chloride into a typical two-dimensional diffusion mode. The two-dimensional chloride diffusion effect in the area around the crack becomes more significant with the increase in crack width. However, the comparison of the test results in Figure 7a,b indicates that there is no significant difference in the chloride concentration variation with the depth at section 1-1 and section 2-2 of the base slab specimens. It should be noted that the influence depth of chloride erosion in the base slab specimen is larger than 30 mm, and the effect of two-dimensional chloride diffusion also covers the section 2-2, which is 30 mm away from the cracked section 1-1.

These phenomena observed from the test results in this paper are in good agreement with the results of related references [12,25,52].

### 3.2. Chloride Diffusion Coefficient

The apparent chloride diffusion coefficients in concrete at different erosion stages were deduced by data fitting with reference to the experimental data processing method of JGJ/T 322-2013 [50] and ASTM C1556-11a (2016) [53]. The chloride diffusion coefficient D (m^2^/s) and the surface chloride concentration $C_{S}$ (% concrete mass) at section 1-1 of the specimen were fitted by MATLAB R2016a software based on measured chloride concentrations given in Figure 6 and Figure 7. According to Fick’s second law, where the influence of the convection zone is considered [11], the chloride diffusion coefficient and surface chloride concentration of concrete were fitted by
(2)$$C(x,t)=C_{0}+(C_{S}-C_{0})\left[1-\mathrm{erf}\left(\frac{x}{2\sqrt{Dt}}\right)\right],$$
where C(x,t) is chloride concentration at x from the concrete surface at the time of t; $C_{0}$ is the initial chloride concentration; $C_{S}$ is the surface chloride concentration; Δx is the depth of the convection zone in the concrete cover; D is the apparent chloride diffusion coefficient of concrete; t is the time; erf( ) is the error function.

The boundary condition is $C(0,t)=C_{S}$, and the initial condition is C(x,0)=0. The fitting results of $C_{S}$ and D are shown in Figure 8.

The apparent chloride diffusion coefficient *D,* shown in Figure 8, represents the average result of chloride corrosion at a certain stage. From the fitting results shown in Figure 8a, it can be seen that the apparent chloride diffusion coefficient of concrete at cracked section 1-1 tends to increase with the increase of crack width in track slab and base slab specimens. The chloride diffusion coefficient in concrete at the cracked section with a maximum crack width of 0.6 mm is 11 to 12 times higher than that at the sound section in track slab specimens, while the corresponding value decreases to 8.5 to 11 times in base slab specimens. It indicates that transverse cracks significantly influence the apparent chloride diffusion coefficient in the concrete at the cracked section. It can be attributed to the existence of a macro transverse crack providing a shortcut for chloride from the side wall of the crack, which greatly accelerates the chloride transmission rate in concrete adjacent to the crack by a typical two-dimensional diffusion mechanism. However, the adverse influence of crack seems to gradually decrease with the increase of drying-wetting cycles. This phenomenon could be understood as the gradual hydration of cement during drying-wetting cycles leading to a more compact structure of concrete, and the subsequent hydration product of cement and dust could infill the crack.

It can be seen from Figure 8b that the surface chloride concentration of the concrete specimen increases with the increase of drying-wetting cycles, and the accumulation rate of chloride ions deposited on the surface of the specimen is higher than the transmission rate to the interior of concrete. The surface chloride concentration of the base slab specimen at the cracked section increases with the increase of maximum crack width, and the influence of crack width gradually decreases from 0 mm to 0.6 mm. It indicates that the existence of a transverse macro crack provides special space for chloride ions deposition. The surface chloride concentration of the track slab specimen is significantly higher than that of the base slab specimen. It can be attributed to the details of the V-shaped groove in the track slab specimen, which could collect chloride solution during the drying-wetting cycles and cause the surface chloride concentration at the bottom of the groove to be higher than that on the top surface.

### 3.3. Transport Model of Chloride in Cracked Concrete

The previous researches [54,55,56] and experimental results in this study indicate that the macro transverse cracks can greatly accelerate the chloride diffusion rate in concrete adjacent to the crack. The mechanism can explain that a macro transverse crack provides a special shortcut for chloride from the side wall of the crack, which changes the one-dimensional diffusion mode in sound concrete into a localized two-dimensional diffusion mode in cracked concrete. As shown in Figure 9, the dual porosity medium model is used to analyze the transmission process of chloride ions in the cracked concrete of the track slab and base slab.

In order to consider the influence of the transverse crack on the localized chloride transmission rate in concrete, a degradation function related to the crack width *f*(*w*) is introduced into the diffusion coefficient model of concrete based on Fick’s second law. The surface chloride concentration, binding effect of chloride, time-varying effect of chloride diffusion coefficient, temperature, relative humidity, and crack width were comprehensively considered in the proposed model. The equivalent diffusion coefficient of concrete at the cracked section can be expressed as:(3)$$D_{\mathrm{cr}}=D_{\mathrm{app}}f(w)=D_{0}k_{R}k_{t}k_{T}k_{H}f(w)$$
where $D_{\mathrm{cr}}$ is the equivalent diffusion coefficient at the cracked section; $D_{\mathrm{app}}$ is the modified apparent chloride diffusion coefficient in sound concrete; f(w) is the degradation function related to the maximum crack width *w*; $D_{0}$ (m^2^/s) is the chloride diffusion coefficient of concrete at the age of 28 days in the standard curing environment (i.e., at a temperature of 20 ± 2 °C and relative humidity of 95%) that can be written as [57]:(4)$$D_{0}=10^{\left(-12.06+2.4w/c\right)}$$
where *w/c* is the water-cement ratio.

$k_{R}$ is the coefficient that accounts for the chloride binding effect on the chloride diffusion coefficient of concrete. According to the research of Martín-Pérez et al. [37], $k_{R}$ can be expressed as follows:(5)$$k_{R}=1/\left(1+\frac{\partial C_{b}}{\omega_{e}\partial C_{f}}\right)$$
where $w_{e}$ is the content of evaporable water, and $\partial C_{b}/\partial C_{f}$ represents the chloride ion binding capacity of the cementitious material, which is the slope of the relation curve between bound chloride ion concentration and free chloride ion concentration in concrete.

$k_{t}$ is the influence coefficient of exposure time on the chloride diffusion coefficient of concrete, and it is generally expressed as power law considering time-exponent *m* [38], which can be expressed as follows:(6)$$k_{t}={\left(t_{0}/t\right)}^{m}$$
where $t_{0}$ is the reference time, usually taken as 28 days; *t* is the exposure time; *m* is the time-dependent constant which depends on the mix proportions of concrete.

$k_{T}$ is the influence coefficient of exposure temperature on the chloride diffusion coefficient of concrete, and according to Arrhenius’ law, it can be written as [57]:(7)$$k_{T}=\left(\frac{U}{R}\left(\frac{1}{T_{\mathrm{ref}}}-\frac{1}{T}\right)\right)$$
where *U* is the activation energy of the diffusion process, which is related to the water-cement ratio of concrete. The *U* values of concrete specimens made of ordinary Portland cement are 41.8 ± 4.0 (kJ/mol) for a *w/c* of 0.4, 44.6 ± 4.3 (kJ/mol) for a *w*/*c* of 0.5 and 32.0 ± 2.4 (kJ/mol) for a *w*/*c* of 0.6 [36]; $T_{\mathrm{ref}}$ is usually taken as 293 K (20 °C).

$k_{H}$ is the influence coefficient of relative humidity on the chloride diffusion coefficient of concrete, according to the research of Ayman et al. [36]. Based on the model proposed by Bažant and Najjar [58], it can be expressed as follows:(8)$$k_{H}={\left[1+\frac{{\left(1-H\right)}^{4}}{{\left(1-H_{C}\right)}^{4}}\right]}^{-1}$$
where $H_{C}$ is the critical relative humidity (usually taken as $H_{C}=0.75$); H is the relative humidity inside the concrete.

According to related research, it was informed that the averaged chloride diffusion coefficient of concrete at the cracked section has a quadratic function or cubic function to the crack width in a non-steady state or partially saturated condition [17,25,34]. Based on the test data of chloride erosion drying-wetting cycles presented in this study, the influence of crack width on concrete’s averaged chloride diffusion coefficient was quantitatively analyzed. A coefficient of degradation effect is introduced to describe the influence of the crack in this study, which is defined as the ratio of the chloride diffusion coefficient in cracked concrete to that in sound concrete. The mean values of the degradation effect coefficients obtained from the track slab specimens and base slab specimens are shown in Figure 10 as scatter plots. The relationship between the degradation effect coefficient and crack width can be obtained by regression analysis of test data adopting different types of functions through MATLAB R2016a software, as shown in Figure 10.

The regression analysis results shown in Figure 10 indicate that the mean values of the degradation effect coefficient at the cracked section of the specimen are generally well-fitted with a cubic function, quadratic function, and exponential function. The fitting accuracy of different functions is in the order of the cubic function, the quadratic function, and the exponential function. In the fitted curves of the track slab and base slab, the correlation coefficients of the quadratic function and cubic function are 0.984 and 0.973, 0.999 and 0.998, respectively. The degradation effect coefficient of crack width on the chloride diffusion in the track slab and base slab specimen can be expressed by cubic function or quadratic function.

It should be noted that the quadratic function models proposed by Kwon [17] and Zhang [25] are also given in Figure 10 for comparison. It is evident that the test result obviously deviates from the quadratic curve of the model proposed by Kwon and Zhang. The quadratic function obtained by regression analysis in this study is consistent with Kwon’s model and Zhang’s model in the form, while it has different coefficients for the quadratic term and primary term. The different coefficients could be attributed to the differences in mix ratio, curing conditions, and exposure conditions.

## 4. Conclusions

Drying-wetting cycle chloride erosion tests were carried out on reinforced concrete track slab and base slab specimens in this study, and the influence of transverse crack on chloride ingress of slab track structure in the coastal environment was analyzed. The following conclusions can be drawn according to the analysis of the experimental results:(1)The experimental results show that the chloride ion concentration of the concrete slab track at the crack section is much higher than that at the intact section, and it increases with the increase of crack width in a range of 0.2 mm to 0.6 mm.(2)The variation of chloride concentration with the depth in the concrete slab track obviously deviates from the chloride diffusion model in the cracked section. The presence and development of transverse cracks in the concrete track slab and base slab could change the one-dimensional diffusion-dominated transmission mode of chloride into a typical two-dimensional diffusion mode.(3)The chloride diffusion coefficient of the concrete slab track at the cracked section is much higher than that at the uncracked section. When the surface crack width reaches 0.6 mm, the apparent chloride diffusion coefficient increases by 11 to 12 times and 8.5 to 11 times for the track slab and base slab specimens, respectively.(4)The surface chloride concentration of the concrete slab track at the cracked section increases with the increase of maximum crack width. Transverse cracks and V-shaped grooves in the CRTS II slab track structure could collect chloride solution during the drying-wetting cycles and provide special space for chloride ions deposition.(5)A degradation effect function *f*(*w*) of the concrete slab track is introduced in the equivalent diffusion coefficient model to express the influence of crack width on the localized chloride transmission rate at the cracked section. The relationship between the degradation effect coefficient of chloride diffusion and the crack width conforms to a cubic function or quadratic function for the track slab and base slab.

## Figures and Tables

**Figure 1 materials-16-03524-f001:**
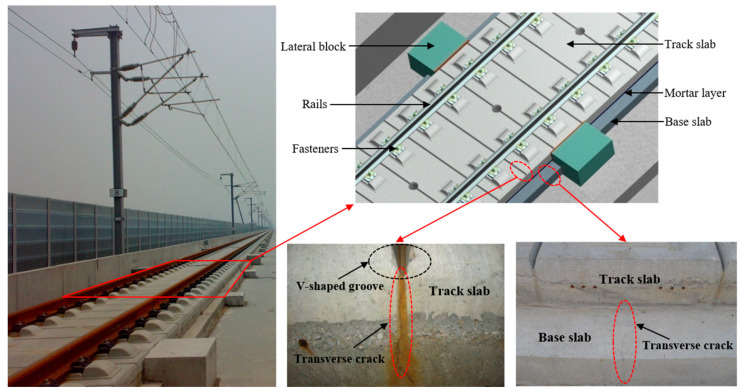
CRTS II slab track structure and transverse cracks.

**Figure 2 materials-16-03524-f002:**
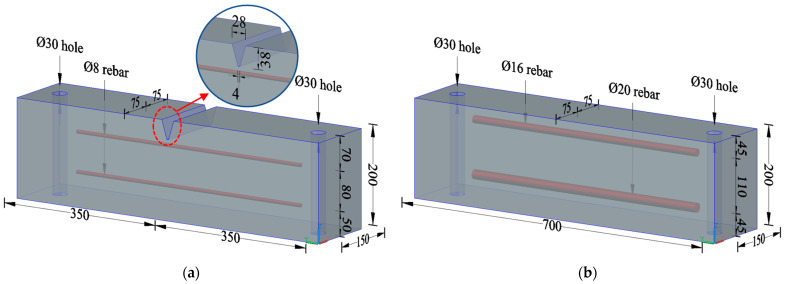
Design of test specimens (all dimensions are in mm). (**a**) Track slab specimen with V-shaped groove; (**b**) base slab specimen.

**Figure 3 materials-16-03524-f003:**
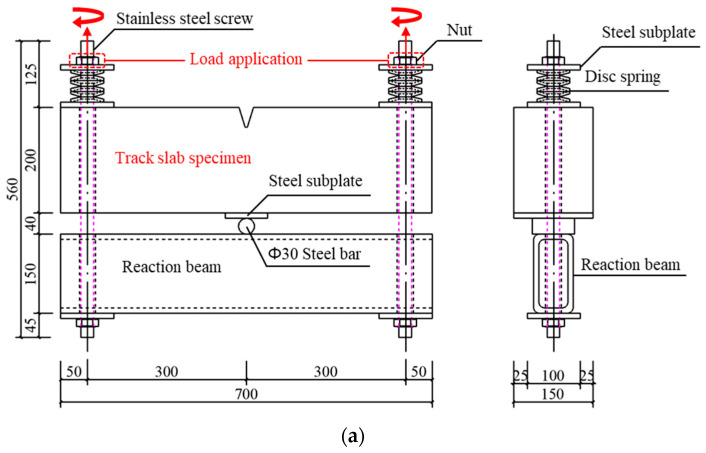
The setups of the loading test and crack measurement. (**a**) Loading system of track slab specimen (mm); (**b**) loading system of base slab specimen (mm); (**c**) loading and strain acquisition; (**d**) crack width observation.

**Figure 4 materials-16-03524-f004:**
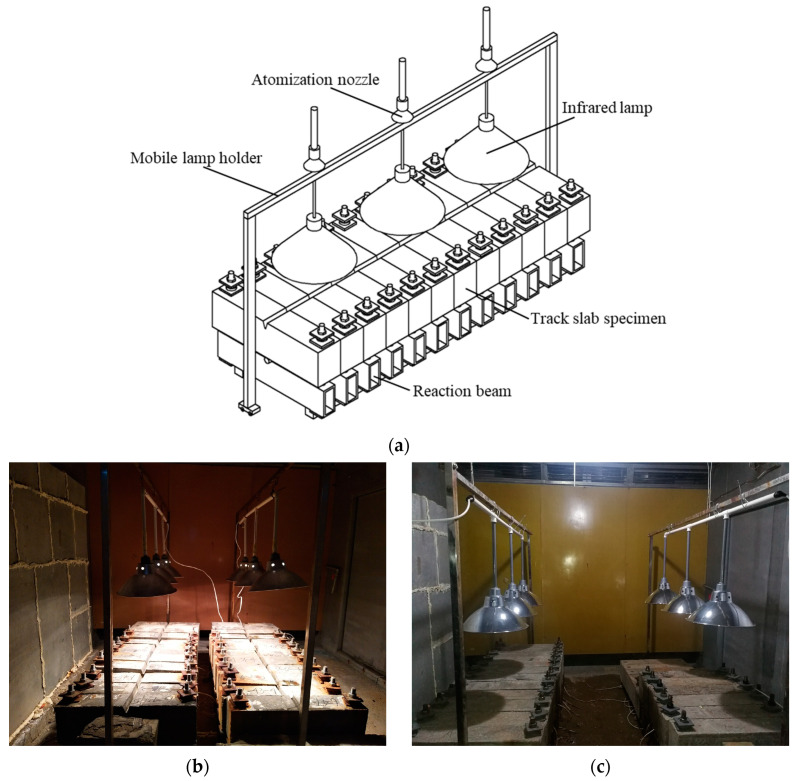
Dry-wetting cycles test device for chloride corrosion. (**a**) Test device for drying-wetting cycles.; (**b**) infrared lamp irradiation; (**c**) standing and ventilated.

**Figure 5 materials-16-03524-f005:**
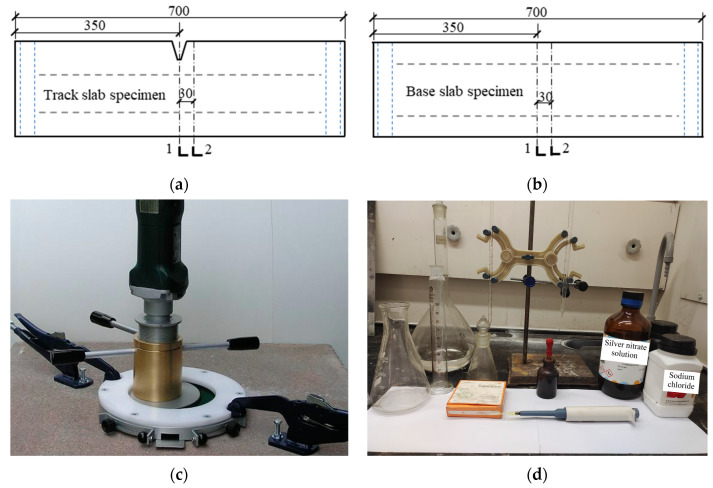
Sampling and determination of free chloride ions concentration (all dimensions in mm). (**a**) Sampling sections of track slab specimen; (**b**) sampling sections of base slab specimen; (**c**) PF-1100 profile grinding machine; (**d**) determination of chloride ions concentration.

**Figure 6 materials-16-03524-f006:**
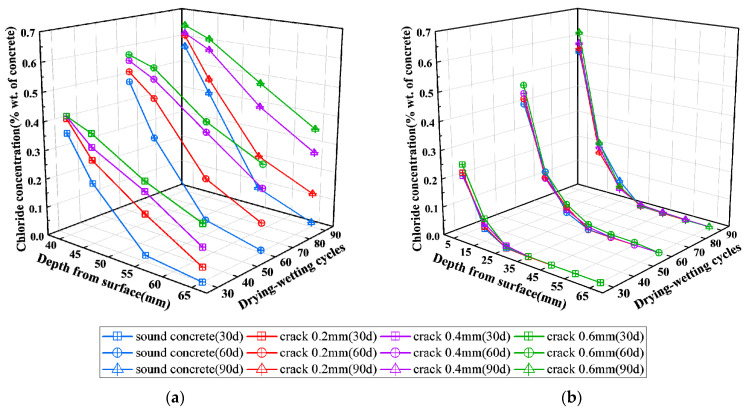
Chloride ions concentration in the concrete of the track slab. (**a**) Cracked section 1-1; (**b**) Uncracked section 2-2.

**Figure 7 materials-16-03524-f007:**
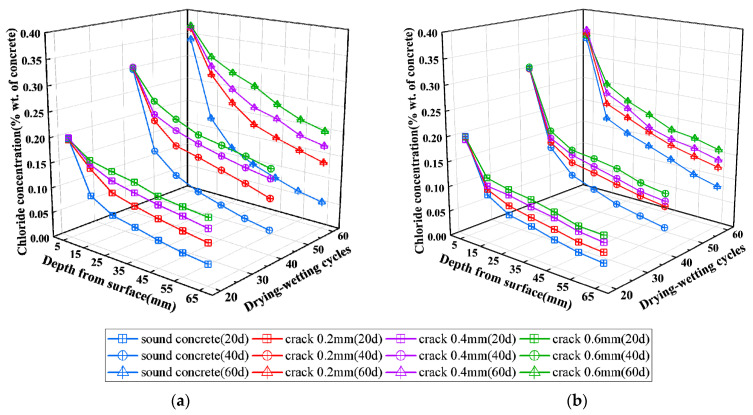
Chloride ions concentration in the concrete of the base slab. (**a**) Cracked section 1-1; (**b**) Uncracked section 2-2.

**Figure 8 materials-16-03524-f008:**
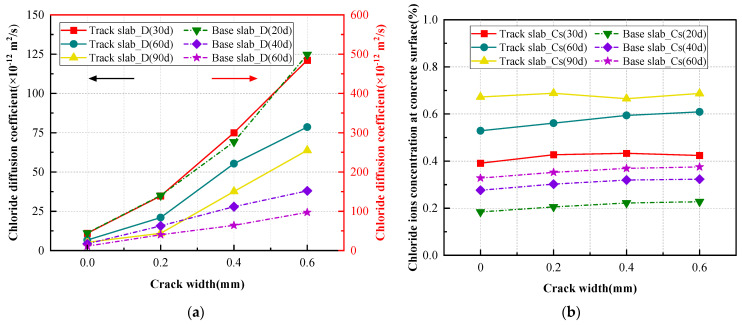
The fitting of D and $C_{S}$ at section 1-1 with different drying-wetting cycles of chlorine corrosion. (**a**) Apparent chloride diffusion coefficient D; (**b**) surface chloride concentration $C_{S}$.

**Figure 9 materials-16-03524-f009:**
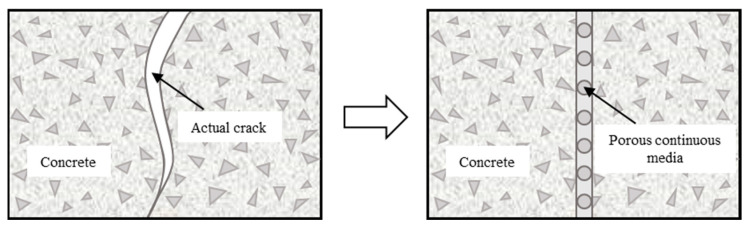
Principle of dual porosity medium model.

**Figure 10 materials-16-03524-f010:**
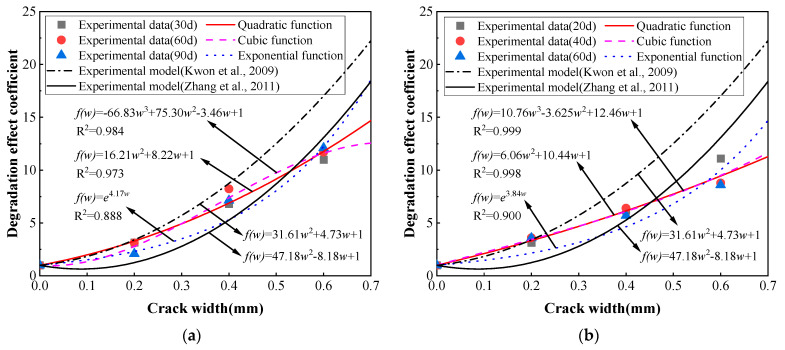
Fitting results of the degradation effect coefficient to crack width with different type functions. (**a**) Track slab specimen; (**b**) base slab specimen [17,25].

**Table 1 materials-16-03524-t001:** Concrete mix proportions and physical properties.

Specimens	Strength Grade of Concrete	W/C ^1^	Cement (kg/m^3^)	Fly Ash (kg/m^3^)	Fine Aggregate (kg/m^3^)	Coarse Aggregate (kg/m^3^)	Water (kg/m^3^)	Water Reducer (kg/m^3^)	28-Day Cube Compressive Strength (MPa)
Track slab	C55	0.31	430	48	680	1157	133	6.4	59.4
Base slab	C30	0.44	273	117	880	954	172	3.9	33.6

^1^ W/C was the water-cement ratio.

**Table 2 materials-16-03524-t002:** Mineral ingredients of Portland cement.

Mineral Ingredient	CaO	SiO_2_	Al_2_O_3_	Fe_2_O_3_	MgO	SO_3_	Insoluble Substance
Content (%)	63.51	22.13	5.63	4.03	1.69	2.57	1.92

**Table 3 materials-16-03524-t003:** Properties and maximum crack widths of the tested specimens.

Type	Specimens ID	Designed Maximum Crack Widths (mm)	Measured Maximum Crack Widths (mm) *	Average Widths (mm)	Loaded/Unloaded
Track slab	A-1~6	0	Sound concrete	0	Unloaded
	B-1~6	0.20	0.21/0.22/0.19/0.21/0.20/0.21	0.21	Loaded
	C-1~6	0.40	0.43/0.41/0.40/0.41/0.39/0.41	0.41	Loaded
	D-1~6	0.60	0.61/0.62/0.61/0.65/0.61/0.62	0.62	Loaded
Base slab	A’-1~6	0	Sound concrete	0	Unloaded
	B’-1~6	0.20	0.21/0.19/0.21/0.21/0.22/0.20	0.21	Loaded
	C’-1~6	0.40	0.40/0.40/0.41/0.42/0.43/0.39	0.41	Loaded
	D’-1~6	0.60	0.60/0.62/0.61/0.59/0.61/0.63	0.61	Loaded

* The specimens were subjected to continue loading when the maximum crack widths were measured.

## Data Availability

Data are contained within the article.

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
