# Peer review of "Experimental Study on the Influence of Transverse Crack on Chloride Ingress in Concrete Slab Track of High-Speed Railway"

_materials, 2023, doi:10.3390/ma16093524_

Round 1

Reviewer 1 Report

In my opinion, the article is an interesting research work. Great effort in conducting experiments. Very well done analysis and description of the experiments. The work focuses on the effect of chloride on erosion, but other factors may also be considered. What effect does the ambient temperature have on the propagation rate of this corrosion?

Author Response

Reviewer’s comments and response letter

Dear Editor and reviewers,

Thank you for your letters dated on 4-Apr-2023 and 6-Apr-2023. We were pleased to know that our work was rated as potentially acceptable for publication in Materials, subjected to adequate revision. We thank the reviewers for the time and effort that they have put into reviewing the previous version of the manuscript. Their suggestions have enabled us to improve our work. Based on the instructions provided in your letter, we have uploaded the revised manuscript. Accordingly, we have highlighted all the changes by red colored text in MS Word.

Appended to this letter is our point-by-point response to the comments raised by the reviewers. The comments are reproduced and our responses are given directly afterward in a different color (red).

We also would like to thank you for allowing us to resubmit a revised version of the manuscript.

We hope that the revised manuscript is acceptable for publication in Materials.

Sincerely,

Xiaochun LIU

Reviewer #1:

Comments and Suggestions for Authors

  1. In my opinion, the article is an interesting research work. Great effort in conducting experiments. Very well done analysis and description of the experiments. The work focuses on the effect of chloride on erosion, but other factors may also be considered. What effect does the ambient temperature have on the propagation rate of this corrosion?

Response: Accepted.

We thank the reviewer for the positive comments on the value of this work and valuable suggestions for the manuscript. The variation of ambient temperature will change the chloride ingress in concrete, the chloride ingress dry-wetting cycle tests designed takes advantage of this characteristic in this study. The ambient temperature for specimens curing and dry-wetting cycle tests were supplemented in the revised manuscript. Moreover, the effect of ambient temperature and relative humidity on the corrosion current density of steel bars embedded in concrete will be reported in subsequent studies. All the revisions can be seen in the revised manuscript, line 142 to line 144 and line 203 to line 205.

Reviewer 2 Report

The study is interesting. However the issue of crack is driven by environmental effects. The literature review should be improved. For example, the effect of thermal-environmental cracks can be found in:

https://doi.org/10.1016/j.engstruct.2023.115789

The authors should further improve the literature review since there are many similar work done by other researchers.

The experimental setup and measurement methods and instrumentation should be fully appended.

The details of specimens used in the tests are missing. There is a need to justify the specimen design and the structural details, materials and design criteria.

SEM results are missing completely. There is a need to obtain SEM data since the topic is related more to materials sciences (not engineering).

Experimental results are insufficient. There is a need to append more measurement results, i.e., exposure class, the transport and porosity, durability, and permeability.

Strength and stiffness reduction characteristics are also missing completely.

The work does not have sufficient experimental data to support the claims.

The manuscript should be proof read by a Native speaker, in order to improve the quality of technical English.

Author Response

Reviewer’s comments and response letter

Dear Editor and reviewers,

Thank you for your letters dated on 4-Apr-2023 and 6-Apr-2023. We were pleased to know that our work was rated as potentially acceptable for publication in Materials, subjected to adequate revision. We thank the reviewers for the time and effort that they have put into reviewing the previous version of the manuscript. Their suggestions have enabled us to improve our work. Based on the instructions provided in your letter, we have uploaded the revised manuscript. Accordingly, we have highlighted all the changes by red colored text in MS Word.

Appended to this letter is our point-by-point response to the comments raised by the reviewers. The comments are reproduced and our responses are given directly afterward in a different color (red).

We also would like to thank you for allowing us to resubmit a revised version of the manuscript.

We hope that the revised manuscript is acceptable for publication in Materials.

Sincerely,

Xiaochun LIU

Reviewer #2:

Comments and Suggestions for Authors

  1. The study is interesting. However the issue of crack is driven by environmental effects. The literature review should be improved. For example, the effect of thermal-environmental cracks can be found in: https://doi.org/10.1016/j.engstruct.2023.115789.

Response: Accepted.

We thank the reviewer for being interested in our work and valuable suggestions for the manuscript. In order to enhance the readability of the article, we have revised the literature review section in accordance with the comments of the reviewers and added relevant references, where the reasons for the appearance of cracks in the CRTS II track structure were described in the introduction. All the revisions can be seen in the revised manuscript, line 34 to line 35.

  1. The authors should further improve the literature review since there are many similar work done by other researchers.

Response: Accepted.

We have further improved the literature review and added the contribution has been made by former researches in accordance with the comment of the reviewer. All the revisions can be seen in the revised manuscript, line 49 to line 60.

  1. The experimental setup and measurement methods and instrumentation should be fully appended.

Response: Accepted.

We have appended experimental setup, measurement methods and instrumentation in the revised manuscript in accordance with the comment of the reviewer. The experimental setup of loading was marked and the local magnification of the strain gauges location on the specimen were added in Fig. 3. In addition, we have added the description to the determination of free chloride ions concentration in concrete.

All the revisions can be seen in the revised manuscript, Fig. 3, Fig. 5 and line 217 to line 222.

  1. The details of specimens used in the tests are missing. There is a need to justify the specimen design and the structural details, materials and design criteria.

Response: Accepted.

We have appended the details of specimens in the revised manuscript in accordance with the comment of the reviewer. The details of the specimens were explained in the “2.1. Design of specimens” section. In order to reflect the details of slab track structure where transverse cracks appeared in the track slab and base slab (as shown in Fig. 1), the test track slab and base slab specimens were designed as segmental model with equal thickness of full-scale track slab and base slab. The materials of concrete and steel reinforcement as well as the structural details including cover thickness of concrete were designed in accordance with the CRTS II slab track structure used in Beijing-Shanghai high speed railway.

All the revisions can be seen in the revised manuscript, Fig. 1 and line 101 to line 111.

  1. SEM results are missing completely. There is a need to obtain SEM data since the topic is related more to materials sciences (not engineering).

Response: Partly Accepted.

SEM is a widely used testing equipment in the field of materials science. SEM can also be used to observe the pore structure and ion transport characteristics of concrete in research. However, in this study, SEM was not used for observation. The main reason is that the test specimen is a reinforced concrete component subjected to continuous bending moment. Cutting and sampling for SEM during the test process will cause damage to the specimen, affecting the stress state of the component and subsequent test progress. The reason of lacking SEM results has been added in the revised manuscript, line 238 to line 244.

  1. Experimental results are insufficient. There is a need to append more measurement results, i.e., exposure class, the transport and porosity, durability, and permeability.

Response: Partly Accepted.

The chloride ingress drying-wetting cycle tests were conducted on reinforced concrete track slab specimens and base slab specimens with different crack widths in this study. The design of specimens and test parameters were based on the practical engineering and relevant specifications. Only the concentration of chloride ions in concrete at different depths was measured by drilling concrete powder samples during the drying-wetting cycle tests. The exposure class is described in in the revised manuscript, line 189 to line 209. The porosity and permeability of concrete is difficult to measure in such a large reinforced concrete slab specimen. Cutting and sampling for microscopic testing will cause damage to the specimen, affecting the stress state of the component and subsequent test progress. The reason of lacking microscopic testing of porosity and permeability results has been added in the revised manuscript, line 238 to line 244. The durability of steel bars embedded in concrete was investigated in this study, and the test results and numerical simulation will be published in the next paper.

  1. Strength and stiffness reduction characteristics are also missing completely.

Response: Partly Accepted.

As to the knowledge of the authors, it is generally accepted that chloride ingress has little effect on the strength and stiffness of concrete member before steel bar lose passivation and begin corrosion [1]. The subsequent corrosion of steel bars will lead to a decrease in the effective sectional area of steel bar, and the corrosion product of the steel bars will cause the concrete to expansive crack and fall off of concrete cover, and the bond behavior between the steel bars and the concrete will decrease, which will lead to reduction of structural strength and stiffness [2].

However, the focus of this manuscript is the effect of transverse cracks on chloride ingress in concrete track structure before severe steel corrosion, so the data of strength and stiffness were not given in this manuscript. In fact, the durability of steel bars embedded in concrete and the reduction characteristics of structural strength and stiffness was investigated in the subsequent study, and the test results and numerical simulation will be published in the next paper.

The reason of lacking strength and stiffness reduction results has been added in the revised manuscript, line 246 to line 248.

[1] Cui, Z.; Alipour, A. Concrete cover cracking and service life prediction of reinforced concrete structures in corrosive environments. Constr. Build. Mater. 2018, 159, 652-671.

[2] Yoon, S.; Wang, K.; Weiss, W.; Shah, S. Interaction Between Loading, Corrosion, and Serviceability of Reinforced Concrete. ACI Struct. J. 2000, 97, 637-644.

  1. The work does not have sufficient experimental data to support the claims.

Response: Partly Accepted.

Due to limitations in research funds, period and experimental condition, we designed and fabricated 48 CRTSII slab track structure specimens with crack widths of 0-0.6 mm. The chloride ions concentration in concrete at different depths of cracked section and uncracked section was tested after chloride ingress drying-wetting cycle tests. The authors believe that experimental data can support conclusions for concrete slab track structure. In addition, the crack effect functions established by previous scholars were added in the revised manuscript for comparison to further verify the reliability of the results.

All the revisions can be seen in the revised manuscript, Fig.10 and line 415 to line 418.

  1. The manuscript should be proof read by a Native speaker, in order to improve the quality of technical English.

Response: Accepted.

We apologized for our poor English. The misspelling and grammatical errors in the manuscript were revised by line-editing and proofreading, and we have tried our best to improve our scientific English writing in the revised manuscript.

Reviewer 3 Report

Questions and suggestions for supplementing the article:

·       Were the assembled experiments performed according to local or international standards? Or were these experiments created on the basis of pre-defined assumptions by the authors?

·       Although I understand why the authors used static loading of the specimens in the experiment, would it be possible to use dynamic low-frequency loading? After all, only the construction of a fixed track is loaded by rail traffic, which would better describe dynamic loading.

·       I recommend better describing the used measuring device and its settings for the measurement needs and more appropriately specifying the strain gauge used, especially what type of strain gauge it is (surface strain sensor?). At the same time, I recommend adding a picture better showing the location of the strain gauges.

·       What other instruments and techniques have the authors used or plan to use in the future to analyse crack behaviour in concrete specimens?

·       What impact will the obtained results have on construction practice? How will the research results affect currently used materials, technology, or the construction of a fixed track?

·       How will the authors continue their research work?

Author Response

Reviewer’s comments and response letter

Dear Editor and reviewers,

Thank you for your letters dated on 4-Apr-2023 and 6-Apr-2023. We were pleased to know that our work was rated as potentially acceptable for publication in Materials, subjected to adequate revision. We thank the reviewers for the time and effort that they have put into reviewing the previous version of the manuscript. Their suggestions have enabled us to improve our work. Based on the instructions provided in your letter, we have uploaded the revised manuscript. Accordingly, we have highlighted all the changes by red colored text in MS Word.

Appended to this letter is our point-by-point response to the comments raised by the reviewers. The comments are reproduced and our responses are given directly afterward in a different color (red).

We also would like to thank you for allowing us to resubmit a revised version of the manuscript.

We hope that the revised manuscript is acceptable for publication in Materials.

Sincerely,

Xiaochun LIU

Reviewer #3:

Comments and Suggestions for Authors

  1. Were the assembled experiments performed according to local or international standards? Or were these experiments created on the basis of pre-defined assumptions by the authors?

Response: Accepted.

The chloride ingress dry-wetting cycle tests were designed according to the “Standard for indoor simulated environmental test method of concrete structural durability”. We have added a supplementary explanation in the revised manuscript in accordance with the comment of the reviewer.

All the revisions can be seen in the revised manuscript, line 199 to line 200.

  1. Although I understand why the authors used static loading of the specimens in the experiment, would it be possible to use dynamic low-frequency loading? After all, only the construction of a fixed track is loaded by rail traffic, which would better describe dynamic loading.

Response: Accepted.

The main focus of this manuscript is to study the chloride ingress in concrete of track structure under specific transverse crack width. A self-balanced loading device as shown in Fig. 3 was designed and fabricated to introduce crack and fulfill the continuous loading during chloride ingress test. The chloride ingress in concrete is a slow process. It should be noticed that the accumulated time of train passage is relatively very short comparing to the service life of track structure. Therefore, the static loading of the specimens was used in the experiment, and the influence of dynamic loading was not considered in this study. However, dynamic low-frequency loading could be used in the subsequent study, which could simulate the opening-closing behavior of transverse crack in concrete track structure in practical engineering.

  1. I recommend better describing the used measuring device and its settings for the measurement needs and more appropriately specifying the strain gauge used, especially what type of strain gauge it is (surface strain sensor?). At the same time, I recommend adding a picture better showing the location of the strain gauges.

Response: Accepted.

The strain gauges used in the loading test were DH1205 surface resistance strain gauges provided by Jiangsu Donghua Testing Technology Co., Ltd. We have added an explanation in line 165. At the same time, the localized magnification of strain gauges arrangement has been added in Fig. 3 in the revised manuscript.

  1. What other instruments and techniques have the authors used or plan to use in the future to analyse crack behaviour in concrete specimens?

Response: Accepted.

The crack observation instrument provided by Beijing ZBL Technology Co., Ltd is used to further observe the development of cracks during the test. Combined with the field measurement and numerical simulation, the crack development process in practical engineering and its influence on the durability of track slab and base slab structure will be analyzed in the subsequent study.

  1. What impact will the obtained results have on construction practice? How will the research results affect currently used materials, technology, or the construction of a fixed track?

Response: Accepted.

The test result indicates that chloride diffusion coefficient of concrete at cracked section is much higher than that at uncracked section. A model of chloride diffusion coefficient of cracked concrete is proposed for concrete slab track based on the experimental results, which could be used to evaluate the durability life of concrete slab track based on chloride ion concentration of concrete in steel bar surface reaches the critical threshold value. The research results can provide theoretical basis for the maintenance of track structure.

All the revisions can be seen in the revised manuscript, line 96 to line 98.

  1. How will the authors continue their research work?

Response: Partly Accepted.

Further research will continue to investigate the subsequent effects of chloride ingress in cracked concrete in concrete slab track: (1) the effect of transverse crack on the corrosion process of steel bar embedded in cracked concrete slab track; (2) the degradation of structural strength and stiffness of concrete slab track; (3) the effect on the durability of the track structure; (4) the effect of durability degradation on the riding comfort and safety of train traffic.

Reviewer 4 Report

C

Author Response

Reviewer’s comments and response letter

Dear Editor and reviewers,

Thank you for your letters dated on 4-Apr-2023 and 6-Apr-2023. We were pleased to know that our work was rated as potentially acceptable for publication in Materials, subjected to adequate revision. We thank the reviewers for the time and effort that they have put into reviewing the previous version of the manuscript. Their suggestions have enabled us to improve our work. Based on the instructions provided in your letter, we have uploaded the revised manuscript. Accordingly, we have highlighted all the changes by red colored text in MS Word.

Appended to this letter is our point-by-point response to the comments raised by the reviewers. The comments are reproduced and our responses are given directly afterward in a different color (red).

We also would like to thank you for allowing us to resubmit a revised version of the manuscript.

We hope that the revised manuscript is acceptable for publication in Materials.

Sincerely,

Xiaochun LIU

Reviewer #4:

Comments and Suggestions for Authors

  1. How many samples according to Fig. 1 was prepare and tested? It would be appropriate to indicate here the designation of the series and the number of samples in each series or at least refer to the fact that it will be given in Table 3.

Response: Accepted.

Two groups of specimens named track slab and base slab were prepared and tested, with 24 specimens in each group, numbered A, B, C and D according to the crack width. We have added the designation of the series and the number of samples in the revised manuscript.

All the revisions can be seen in the revised manuscript, line 109 to line 111.

  1. It should be MPa in Table 1.

Response: Accepted.

We apologized for our carelessness. The misspelling and grammatical errors in the manuscript were revised by line-editing and proofreading, according to the reviewer’s suggestions.

All the revisions can be seen in the revised manuscript.

  1. Why was a different static scheme/different way of loading track slab specimen and base slab specimen chosen? Why was considered only one steel bar in the center of diameter 30mm in the case of track slab specimen and why 2 times steel bar (in thirds) of diameter 30mm in the case of base slab specimen?

Response: Accepted.

Different way of static loading was selected for track slab specimen and base slab specimen due to the different details of track slab and base slab in engineering practice. The center of the track slab specimen was supported by a single steel bar and loaded at both ends to induce cracks at the weakest section of the V-shaped groove. As to the base slab specimen, there is no obvious weak section and transverse crack could appear at any section. Therefore, two steel bars were selected to support and loaded at both ends to induce cracks at the pure bending section in the middle of base slab specimen.

We have added a related explanation in section “2.4. Loading and crack formation” in the revised manuscript.

All the revisions can be seen in the revised manuscript, line 158 to line 159.

  1. It is not clear from the text whether these are crack widths reached during loading (and partially closed and reduced after unloading), or crack widths after unloading, so irreversible (plastic deformation). Or was this entire assembly subjected to dry-wetting cycles?

Response: Accepted.

The specimens were subjected to continue load when the maximum crack widths were reached. We have added an explanation in Table 3. In addition, the entire assembly subjected to dry-wetting cycles as shown in line 187 to line 190 and Fig. 4.

All the revisions can be seen in the revised manuscript, line 179 to line 180.

  1. It should be chapter 3.3 (chapter 3.2. is in line 267), renumber it.

Response: Accepted.

We apologized for our carelessness. We have renumbered the chapter and made revisions according to the reviewer’s suggestions.

All the revisions can be seen in the revised manuscript.

  1. Literature [16] is Martin-Petez et al., please, correct is what is valid.

Response: Accepted.

We have checked and corrected all the literature in the revised manuscript according to the reviewer’s suggestions.

All the revisions can be seen in the revised manuscript.

  1. Literature [33] is not cited in the text (it omits the citation in the text) - is it necessary to mention it? Then you need to complete the citation in the text. If it is not necessary to cite it, it should be deleted from References.

Response: Accepted.

We have carefully checked and corrected all the literature one by one in the revised manuscript according to the reviewer’s suggestions.

All the revisions can be seen in the revised manuscript, line 304.

Round 2

Reviewer 2 Report

The revision has been improved. However, the technical writing is not well structured.

The structural details of the components and specimens are incomplete. The track components and their dynamic properties are still missing.

The conclusion needs to be drawn in conjunction with the structural details and the track components, otherwise the work will be technically flawed.

Author Response

Reviewer #2:

Comments and Suggestions for Authors

  1. The revision has been improved. However, the technical writing is not well structured.

Response: Accepted.

We apologized for our poor English. The misspelling and grammatical errors in the manuscript were checked and corrected by professional line-editing and proofreading, which are completed by native English speakers. We have tried our best to ensure scientific English writing in the revised manuscript and sincerely hope the revision will be satisfactory

  1. The structural details of the components and specimens are incomplete. The track components and their dynamic properties are still missing.

Response: Accepted.

In order to better display and understand the details of the slab track structure and specimens, we have appended a schematic diagram of the actual track structure in Fig. 1 and replaced the three-view drawing of specimens in Fig. 2 with a three-dimensional perspective drawing.

The main focus of this manuscript is to study the chloride ingress in concrete of track structure under specific transverse crack width. The chloride ingress in concrete is a slow process. It should be noticed that the accumulated time of train passage is relatively very short comparing to the service life of track structure, and it only accounts for 0.54 % of the service life of track structure. Therefore, the dynamic properties of track components and the influence of dynamic loading were not considered in this study. The impact of dynamic action effects on crack development in concrete slab track will be considered in our subsequent research.

  1. The conclusion needs to be drawn in conjunction with the structural details and the track components, otherwise the work will be technically flawed.

Response: Accepted.

We have related the conclusion to the structural details and the track components according to the reviewer’s suggestions.

All the revisions can be seen in the revised manuscript, from line 427 to line 447.
